# Prevalence and Factors for Anxiety during the COVID-19 Pandemic among College Students in China

**DOI:** 10.3390/ijerph18094974

**Published:** 2021-05-07

**Authors:** Jing Guan, Cuiping Wu, Dandan Wei, Qingqing Xu, Juan Wang, Hualiang Lin, Chongjian Wang, Zhenxing Mao

**Affiliations:** 1Department of Epidemiology and Biostatistics, College of Public Health, Zhengzhou University, Zhengzhou 450001, China; guanjing@stu.zzu.edu.cn (J.G.); wucuiping@zzu.edu.cn (C.W.); weidandan@gs.zzu.edu.cn (D.W.); 202022272014946@gs.zzu.edu.cn (Q.X.); 202022272014933@gs.zzu.edu.cn (J.W.); tjwcj2008@zzu.edu.cn (C.W.); 2Department of Epidemiology, School of Public Health, Sun Yat Sen University, Guangzhou 510080, China; linhualiang@mail.sysu.edu.cn

**Keywords:** anxiety, COVID-19, China, factors, prevalence, college students

## Abstract

Background: Knowledge of the impact of COVID-19 on the mental health of college students remains limited. Our aim is to investigate the prevalence of anxiety and explore the potential risk and protective factors of anxiety. Methods: A cross-sectional survey was adopted and a total of 24,678 college students were included from Zhengzhou, Henan Province, China, during February, 2020. Anxiety was assessed by using the Generalized Anxiety Disorder tool (GAD-7). Multiple logistic regression models were established for exploring potential factors of anxiety. Results: The overall prevalence of anxiety was 7.3%. After adjusting for potential confounders, sex, place of residence, worried level, fear level, cognitive levels, and behavior status were found to be associated with anxiety (*p* < 0.05). Students with positive preventive behaviors showed a protective effect against the anxiety symptoms compared to those with negative preventive behaviors. In contrast to the high-cognition category, participants at a low cognitive level were 14.9% more likely to present anxiety symptoms. Conclusion: This large-scale study assessed the prevalence of anxiety and its potential influencing factors among college students. It suggests that the government could strengthen health education related to COVID-19 and supervise the performance of preventive behaviors to handle anxiety.

## 1. Introduction

The coronavirus disease 2019 (COVID-19) widely and rapidly spread across national borders and continents, escalating into a global health crisis [1,2]. As the epidemic developed, the World Health Organization (WHO) expressed deep concerns about the severity of the spread of the outbreak, declared it a Public Health Emergency of International Concern (PHEIC) on 30 January 2020, and assessed the new coronary pneumonia as a global pandemic on 11 March 2020. The negative impact on the economy, social activity, and public health, as well as the uncertain treatment and prognosis of the disease, serious shortages of medicine resources, social isolation, and media information overload, all led to an atmosphere of anxiety around the world [3,4]. Meanwhile, college students comprise a population that is considered particularly vulnerable to mental health concerns due to the exposure to multiple stressors unique to this developmental period [5]. Therefore, it can be inferred that the current pandemic situation may cause major impacts on college students. 

Years lived with disability (YLDs), dominates in measuring non-fatal health loss to track progress because the disease burden, the burden attributed to anxiety disorders from 1990 to 2013, relatively increased by 42.1% [6]. Meanwhile, anxiety disorders are the primary cause of non-fatal burden among the population [7]. Excessive anxiety can be detrimental physically and mentally [8], which perhaps reflects in weakening the body’s immune system and then consequently increasing the risk of contracting the virus, being afraid of seeking medical assistance due to regarding hospitals as a source of contagion, influencing the ability to make rational decisions, and impacting normal activities and behaviors [9].

Diseases can trigger psychological symptoms, especially epidemics caused by disease with uncertain pathogenesis and more likely to result in threatening situations [10]. The COVID-19 epidemic has had a significant socio-psychological impact on society and has triggered a wide variety of psychological problems, which mainly concentrated in anxiety, panic disorder, depression, obsessive-compulsive disorder, hostility, psychoticism, PTSD, etc. [11,12]. During epidemics, the number of people whose psychological health is affected exceeds the number of infected [13]. It was reported by a recent meta-analysis that the prevalence of anxiety among the general population during the COVID-19 pandemic reached 13.9% [14], which was higher than the rate of 5% before the outbreak [15]. In addition, it indicated that approximately 24.9% of college respondents in China manifested psychological anxiety symptoms during the COVID-19 outbreak, which was higher than previously reported as well [16]. There is usually a high prevalence of psychological symptoms among college students [17]. It was reported that anxiety continues to be the most common problem in assessments of psychological symptoms among students, which can affect students’ motivation, concentration, and social interactions that are crucial for students to succeed in academia [18].

From the Chinese experience, treating the infected patients aggressively while protecting susceptible populations and cutting off transmission routes proved to be a huge success in the fight against the COVID-19 virus [19]. On no account can the significance of mental health status in addition to clinical physical conditions be underestimated; it poses great challenges on public health services. As is well known, psychological conditions also play a crucial role in effective public health strategies practiced in pandemic control and prevention, such as risk management, vaccination, and hygiene practices, while mental health anxiety is a key factor in influencing the success or failure of these measures [20]. It is high time to examine the situation of anxiety among students and provide scientific guidance in formulating targeted policies in consideration of the grave epidemic situation in China currently. 

Therefore, this study during the peak period of the COVID-19 outbreak aimed to investigate the prevalence of anxiety, understand the cognitive level of COVID-19 among college students, and identify the possible risk and protective factors giving rise to anxiety. This may reveal how individual characteristics, cognition, and preventive behaviors impact anxiety in COVID-19, give assistance to psychological guidance and interventions for college students, and provide a theoretical basis for government agencies in the formulation and implementation of policies. 

## 2. Methods

### 2.1. Study Participants

The cross-sectional survey was adopted to assess worried level, fear level, cognitive level, behavior status, and anxiety in college students during the COVID-19 pandemic by using an anonymous online questionnaire through an online survey platform (“Survey Star”, Changsha Ran Xing Science and Technology, Shanghai, China). A total of 26,377 valid questionnaires were collected, which recruited college students by using a cluster sampling method from Zhengzhou city, Henan Province, China, from 4 February 2020 to 12 February 2020. Participants aged <18 years or aged >25 years or those who took ≤100 s to fully respond to the questions (*n* = 1699) were excluded so as to control quality [21]. In total, 24,678 college students in this analysis met the criteria and were included.

This study was approved by the Ethics Committee of Zhengzhou University. All study participants consented for participation in this study.

### 2.2. Data Collection

The following demographic characteristics and psychosocial factors were collected to develop a standard questionnaire: sex, age, place of residence, the cognition about COVID-19, information sources and access, behavior, mental state (worry, fear, anxiety), and other factors among all participants. Place of residence was divided into 3 categories: city, county-level city, and rural. Both worried and fear levels among college students during the pandemic were divided into 2 levels: high and moderate/low/none. All behaviors (including “Canceling hanging out with friends”, “Wearing a mask”, “A significant increase in the frequency of hand washing”, “Giving up going out to where crowds congregate”, “Dropping the plan of home-returning or travel during Spring festival”, and “Calling off family outings and visiting activities”) were assessed, then the behavior status was defined as positive; as long as there was one behavior unexecuted, the behavior status was defined as negative. Cognitive level was determined by the responses to nine COVID-19 related questions, which contained questions involving “awareness condition”, “learn the epidemic news timely”, “the route of transmission”, “the correct expression of COVID-19”, “infectivity”, “the period of quarantine”, “the typical post-infection symptoms”, “the effective ways of precautions” and “the selection of effective protection masks”. Each answer to the question was assembled to present “true” or “false or don't know”. A correct response was given a score of 1, and an incorrect or “don't know” response was scored 0. The possible total knowledge score ranged from 0 to 9, then the cognition scores were categorized as a score of <6, 6, or >6 based on the median split, which could be divided into low, moderate, and high levels of cognition, along with higher scores representing higher levels of cognition.

The Chinese version of the Generalized Anxiety Disorder tool (GAD-7) was applied to assess anxiety [22]. A GAD-7 score ≥10 was considered as a reasonable cutoff point to screen and identify clinical anxiety cases [21], when the sensitivity and specificity exceeded 0.80 [23]. Meanwhile, it defined four categories: no (0–4), mild (5–9), moderate (10–14), and severe anxiety (≥15), which had high internal consistency and good test–retest reliability to account for levels of anxiety among college students [24].

### 2.3. Statistical Analysis

Categorical data were represented as frequencies (%) and were compared using Pearson’s chi-squared test. Continuous data were presented by means and standard deviations (SD) and compared with Student’s *t*-tests. The logistic regression models were used to estimate odds ratios (ORs) and 95% confidence intervals (CIs). Two models were developed: (1) unadjusted; and (2) adjusted for sex, place of residence, worried level, fear level, cognitive level, and behavior status. All analyses were performed by using SPSS 21.0 for Windows. All statistical tests were two-sided, with *p* < 0.05 considered statistically significant.

## 3. Results

### 3.1. Demographic Characteristics of the Participants

Among 24,678 participants included in the analysis, 55.2% were male participants. The mean age of participants was 20.51 (SD 1.28) years old. As shown by the relationship between the demographic variables of students and anxiety conditions in Table 1, participants with different anxiety conditions varied in sex, place of residence, cognitive level, worried level, fear level, and behavior status (all *p* < 0.05).

### 3.2. Prevalence of Anxiety

The overall anxiety prevalence was 7.3% during the COVID-19 pandemic among college students, while the proportions of moderate and severe anxiety were 5.2% and 2.1%, respectively. The prevalence for women was higher than men (6.9% vs. 7.8%). Figure 1 shows the prevalence of anxiety in participants by place of residence and sex. The highest prevalence of anxiety was 9.9% with women in the city, and 7.8% with men in the city. The lowest prevalence of anxiety was 6.3% with rural men, and 6.8% with rural women. Participants residing in a county-level city had the middle prevalence of anxiety, which was the same condition in both men and women groups. Participants living in a city had the highest prevalence of anxiety symptoms, and participants who lived rurally had the lowest prevalence of anxiety symptoms among students with the same gender.

As can be seen from Figure 2, there was a difference in the prevalence of anxiety symptoms between those who actively performed the certain specific preventive behaviors and who did not, including “Giving up going out to where crowds congregate”, “Canceling hanging out with friends”, “Dropping the plan of home-returning or travel during Spring festival” and “Calling off family outings and visiting activities”. Students who did not possess these behaviors were more prone to anxiety symptoms.

### 3.3. The Cognitive Level about COVID-19

The mean for the total cognition score was 6.32 out of a possible score of 0–9, and the median was 6 (IQR, 6–7). As shown in Table 1, a total of 11,436 (46.3%) were categorized as having high cognition, with a score of >6; 7107 (28.8%) had moderate cognition with a score of 6; and 6135 (24.9%) had low cognition, with a score of <6. A knowledge gap was found in “the route of transmission” (37.9% participants chose the correct answer), “the typical post-infection symptoms” (38.1%), and “the effective ways of precautions” (42.6%) while the accuracy in the other questions was comparatively higher, even as high as 90%, as described in Table 2.

### 3.4. The Positive or Risk Factors of Anxiety

The results indicated that living in county-level city areas, in comparison with rural areas, increased the likelihood of anxiety (OR 1.288 [95% CI 1.140–1.457]), just like living in a city 40.4% increased the likelihood of anxiety (OR 1.404 [95% CI 1.237–1.595]). In contrast to the highest cognition category, participants at the lowest cognitive level were 14.9% more likely to present anxiety symptoms (OR 1.149 [95% CI 1.016–1.300]), while cognition at moderate level had no significant effect on anxiety. Compared with high worried level, moderate/low/no worried level participants had significantly increased odds of anxiety (OR 5.505 [95% CI 4.783–6.337]), just like the high fear level group had an 80.3% increased likelihood (OR 1.803 [95% CI 1.467–2.217]) compared with participants with a moderate/low/no fear level. Students with positive preventing behaviors showed a protective effect against anxiety symptoms compared to those with negative preventive behaviors; students with negative preventive behaviors were 15.9% more likely to present anxiety symptoms (OR 1.596 [95% CI 1.437–1.773]). Detail information is shown in Table 3.

## 4. Discussion

This was a large-scale cross-sectional epidemiological study investigating the prevalence of anxiety symptoms among college students during the COVID-19 pandemic and explored factors associated with anxiety. Our results indicated that the prevalence of anxiety was 7.3%, while 2.1% students experienced severe anxiety and 5.2% experienced moderate anxiety. College students’ anxiety regarding the pandemic was related to their sex, place of residence, the cognitive level about COVID-19, prevention behavior level, and mental state (worried and fear level).

Grave concerns were raised along with the increasing number of cases and widening geographical spread of the disease, which brought out several sources of stressors that led to college students’ anxiety about COVID-19. Additionally, this is consistent with the evaluated situation of psychological symptoms among populations during previous epidemics such as SARS [25]. The sources of stressors included, on the one hand, threats to the health of individuals and their relatives, the afraid atmosphere of suspected exposure and infection opportunities, as well as the development of the unknown virus [26]. Additionally, anxiety symptoms might have been related to the impact of the diseases on their academic progress and performance, future employment [25], and challenges of remote learning. Meanwhile, in order to prevent further disease transmission, measures such as the suspension of public transportation, closure of gathering places, restrictions to travel (even lockdown), quarantine [3], and vigorous epidemic surveillance, all caused physical distancing and the lack of interpersonal communication [27], resulting in there being no way to release harmful emotions, which resulted in deteriorating anxiety situations to a large extent [28]. It has even been reported that 45% of students have probable acute stress, anxiety, or depressive symptoms during the COVID-19 pandemic [29], and student status was associated with a greater psychological impact of the outbreak and higher levels of stress, anxiety, and depression [25]. Given the severe condition, it was necessary to assess the psychological status of college students during COVID-19 and identify related factors to take pointed measures.

Results suggested that living in rural areas was conducive to presenting lower anxiety prevalence among college students compared with those living in a city or county-level city, unlike the indications which other studies have put forward that higher anxiety levels were noted for students living in rural areas [20]. This discrepancy could be explained by several reasons. Firstly, there was probably no significant distinction in the distribution of sanitary resources and preventive strategies between the areas of respondents. Thus, the larger mobility of mass migration movement, higher population density, and closer interactions in the city and county-level city, all provided likelihood for accelerating the spread of the virus and raising the probability of contact with the pathogen. On the other hand, the emergence and outbreak of infection was first discovered in a city, and the prevalence was even higher and grew faster in urban areas during the preliminary stages, which caused the spread of anxiety emotions among respondents in cities. Meanwhile, whatever the place of residence was, women presented a higher prevalence of anxiety than men. This was in line with other studies which indicated that women’s psychological coping and adjustment ability is lower than men’s in the face of major stress events such as public health emergencies, which could be interpreted as women’s more sensitive characters in general [30,31].

The analysis result was in accordance with previous studies that receiving more relevant knowledge was correlated with lower levels of psychological anxiety. It reported that knowledge and guidance about preventive behaviors made positive contributions to mitigating the spread of and exaggerated or immoderate anxiety due to COVID-19 [32]. As part of health risk communication, knowledge related to the pandemic situation and correct preventive measures might significantly predicted health anxiety status among college students [33]. Students in a low cognitive level tended to be entangled in and even be credulous to the “infodemics” [34], such as fabricated information, “fake news”, and conspiracy theories distributed through news and social media platforms [35], which make it difficult to find information that is timely, trustworthy and accurate; therefore, they are susceptible to speeches stirring up emotion when browsing the Internet which trigger anxiety symptoms. Accurate knowledge helped individuals respond to and defeat the outbreak with a positive attitude and informed positive feedback [25], in line with results from another study [36], where higher COVID-19 knowledge scores based on a comprehensive understanding of pandemic lowered the likelihood of occurrence of negative emotion events and potentially dangerous and non-standard preventive practices; anxiety symptoms could be relieved.

The study also suggested that positive preventing behaviors might be a protective effect against anxiety symptoms, exactly as the lessons learned from the SARS outbreak in 2003 suggested [37]. People’s adherence to prevention measures, which reflected practice situations (part of the KAP theory), can further promote the implementation of strategies that prevent the spread of epidemics from worsening [38]. Consistent with studies conducted elsewhere, nonstandard implementation of preventive behaviors (such as wearing masks incorrectly or choosing ineffective masks) can attribute to increasing anxiety during the COVID-19 outbreak [39]; meanwhile, the formation of hand hygiene practice was associated with lower DASS-21 anxiety scores in Polish and Chinese respondents [40]. There was also an investigation which verified that persons who were highly compliant with prevention measures such as quarantine performed better in cognition tests toward COVID-19, and these two factors, cognitive level and behavior status, were associated with optimistic psychological outcomes [41]. It could be interpreted that thorough preventive behaviors lay a foundation of confidence, combatting against the virus and reducing the risk of adverse psychological outcomes by means of positive psychological suggestion rather than a so-called false sense of security [25]; meanwhile, most universal, selective, and indicated prevention programs may contribute to reducing symptoms of anxiety [42]. Additionally, observing measures such as travel restrictions and wearing masks minimized the chance of contact with clinical or suspected cases and cut off transmission, which alleviated the anxiety of being contaminated. Psychological and physiological mechanisms are activated under COVID-19 epidemic threat, such as eye blink rates, breathing patterns, and humoral factors, which indicate that stress status could fluctuate [43]. The incidence of raised stress-related factors along with the development of COVID-19 can, to some extent, contribute to mental health disorders. It could be inferred that the level of cognition of COVID-19 and preventing behaviors might influence the incidence of anxiety by means of affecting stress-related factors.

Our study has several strengths. Firstly, as far as we know, it contained a large sample size to assess the prevalence of anxiety. Secondly, anxiety condition was certificated using a standardized questionnaire (GAD-7). Thirdly, we screened the participants in accordance with the requirements of this study to further guarantee the reality. Finally, our results inform people vulnerable to anxiety and indicate the factors related to psychosocial symptoms.

Nevertheless, some limitations should be considered. Firstly, although models were adjusted for many important covariates, some possible residual confounding factors may remain. Further research is needed to evaluate those relationships and verify the stability of these results. Secondly, other demographic characteristics statistics such as specific academic year, major, and income were not available; therefore, we could not analyze how student mental health problems differed by these factors. Thirdly, the survey did not evaluate the influence of other recent life events which could have caused anxiety and affected the assessment of comorbidity. Finally, the study was designed as a cross-sectional survey, which does not establish a causal hypothesis.

## 5. Conclusions

The psychological conditions, especially the prevalence of anxiety, were not at an optimistic level during the COVID-19 pandemic among college students; individuals living in a city or county-level city area were a relatively vulnerable group. High cognitive levels to COVID-19 and positive preventive behavior status were closely associated with a low possibility of preventing anxiety symptoms and positive mental health outcomes. It suggests that carrying out promotions and health education related to COVID-19, ensuring the timeliness, authority, and accuracy of information, may be regarded as effective measures to raise the cognitive level up to a higher level so as to partially manage anxiety. In addition, supervising and urging preventive behaviors plays a role in maintaining positive behavior status. Meanwhile, giving targeted assistance to vulnerable groups and monitoring should also be taken into consideration.

## Figures and Tables

**Figure 1 ijerph-18-04974-f001:**
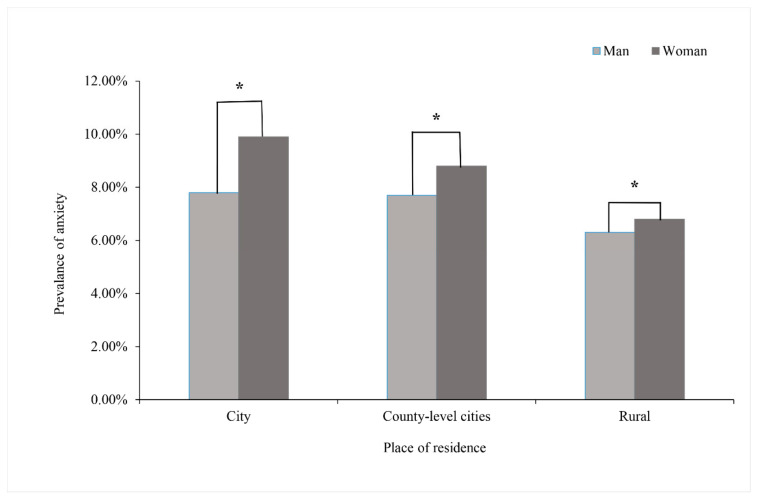
The prevalence of anxiety symptoms in participants by place of residence and men compared with women, * *p* < 0.05.

**Figure 2 ijerph-18-04974-f002:**
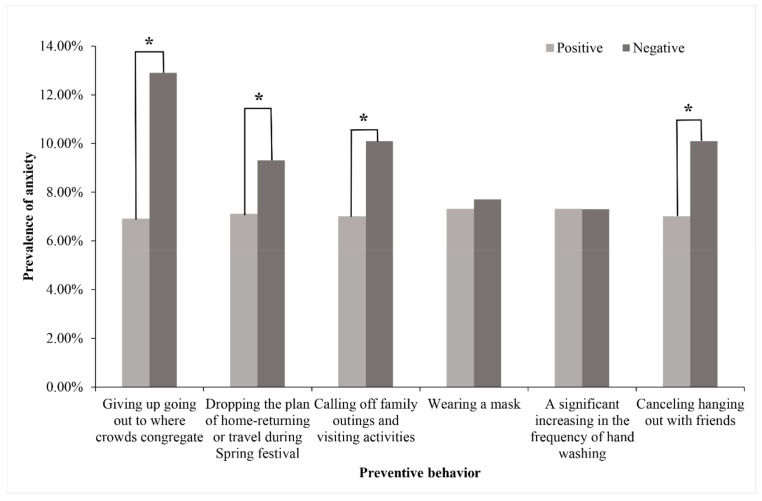
The prevalence of anxiety symptoms classified by preventive behaviors. Performed group compared with not performed group, * *p* < 0.05.

**Table 1 ijerph-18-04974-t001:** Characteristics of the study participants by anxiety status.

Characteristics	All Participants *n* = 24,678	No Anxiety *n* = 22,876	Anxiety *n* = 1802
Sex (%)			
Man	13,630 (55.2)	12,693 (55.5)	937 (52.0)
Woman	11,048 (44.8)	10,183 (44.5)	865 (48.0)
Place of resident (%)			
City	4360 (17.7)	3974 (17.4)	386 (21.5)
Rural	5063 (20.5)	4648 (20.3)	415 (23.0)
County-level city	15,255 (61.7)	14,254 (62.3)	1001 (55.5)
Worried level (%)			
High	18,012 (73.0)	16,340 (71.4)	1672 (92.8)
Moderate/Low/None	6666 (27.0)	6536 (28.6)	130 (7.2)
Fear level (%)			
High	10,796 (43.7)	9313 (40.7)	1483 (82.3)
Moderate/Low/None	13,882 (56.3)	13,563 (59.3)	319 (17.7)
Cognition level (%)			
High	11,436 (46.3)	11,436 (46.3)	783 (43.5)
Moderate	7107 (28.8)	6566 (28.7)	541 (30.0)
Low	6135 (24.9)	5657 (24.7)	478 (26.5)
Behavior Status (%)			
Negative	6432 (26.0)	5810 (25.4)	622 (34.5)
Positive	18,246 (74.0)	17,066 (74.6)	1180 (65.5)

Data are presented as the mean (SD) normal distribution of continuous variables and numbers (percentages) for categorical variables; *p*-values were calculated using Student’s *t*-test and chi-squared. Compared with No anxiety, *p* < 0.05.

**Table 2 ijerph-18-04974-t002:** Frequency distribution of the responses to questions of cognition to COVID-19.

Questions		*n* (%) of Correct Responses
Knowledge about COVID-19		
Q1	Awareness condition	24,580 (99.6%)
Q2	Timely learning of epidemic news	24,657 (99.9%)
Q3	The route of transmission	24,550 (99.3%)
Q4	The correct expression of COVID-19	15,124 (61.3%)
Q5	Infectivity	9363 (37.9%)
Q6	The period of quarantine	21,586 (87.5%)
Q7	The typical post-infection symptoms	9414 (38.1%)
Q8	The effective precautions	10,520 (42.6%)
Q9	The selection of effective protection masks	16,324 (66.1%)

**Table 3 ijerph-18-04974-t003:** Independent association of characteristics of study participants and anxiety during the COVID-19 epidemic in China.

Characteristics	All Participants OR (95%CI)	
	Model 1	Model 2
Sex		
Women	1.00 (ref)	1.00 (ref)
Men	**0.869 (0.789, 0.957)**	1.051 (0.951, 1.162)
Place of residence		
Rural	1.00 (ref)	1.00 (ref)
County-level city	**1.271 (1.129, 1.432)**	**1.288 (1.140, 1.457)**
City	**1.383 (1.224, 1.563)**	**1.404 (1.237, 1.595)**
Worried level		
Moderate/Low/None	1.00 (ref)	1.00 (ref)
High	**5.145 (4.294, 6.164)**	**1.803 (1.467, 2.217)**
Fear level		
Moderate/Low/None	1.00(ref)	1.00 (ref)
High	**6.770 (5.932, 7.663)**	**5.505 (4.783, 6.337)**
Cognition level		
High	1.00 (ref)	1.00 (ref)
Moderate	**1.150 (1.021, 1.294)**	1.104 (0.982, 1.242)
Low	**1.121 (1.000, 1.256)**	**1.149 (1.016, 1.300)**
Behavior status		
Positive	1.00 (ref)	1.00 (ref)
Negative	**1.548 (1.399, 1.714)**	**1.596 (1.437, 1.773)**

Abbreviations: CI, confidence interval; OR, odds ratio. Model 1, unadjusted. Model 2, adjusted for sex, place of resident, cognition level, worried level, fear level, behavior status. Bold: *p* < 0.05

## Data Availability

The data are available from the corresponding author on reasonable request.

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
