# Peer review of "Prevalence and Factors for Anxiety during the COVID-19 Pandemic among College Students in China"

_ijerph, 2021, doi:10.3390/ijerph18094974_

Round 1

Reviewer 1 Report

The manuscript is nicely organised. The statement of the problem is clearly justified. Objectives are clearly investigating. Methods are appropriate. Results and conclusion are well read. However, some points are not clearly identified. I have listed my observation below and hope this will help the author in improving capital of this manuscript. I also recommend minor revision.

Abstract

Line 11-17 is not clearly purpose and method of the study.

Line 22-25 should be added more detailed of practical implications.

Intro

Line 29-37 should be revised why COVID-19 pandemic is generally impacted of anxiety, especially university students.

Line 62-69 should be clear why are Chinese student?

Line 70-79 should be a clear objective of the study.

Methods

The sample size should be clear why and how did you selected 1,699 from 24,678 students. What calculations are you using to select these samples? What time/date/year?

Results

Read well and clear presentation.

Discussion

Elaborate

Conclusion

This section should be clear what main findings, added more detailed how and what practical implications.

Reviewer 2 Report

This study stresess factors affecting college students (18-25 years old) in a region of china. The following issues need to be addressed:

-Please provide a copy of the survey administered. It is unclear how many items comprosed the survey, how the authors evaluated the state of anxiety and behavioral status. It is unclear whether well known psychological questionnares were used for asessment of such aspects. If not this represents a major limitation of the study.

-Are all responders having the same cultural background? please clarify.

-Is there any information about effect of financial issues of responders included in the survey?

-Please stats if the data gathered by the survey will be made publically available to orther researchers.

-Please specify the indepoendent and dependent variables in the logistic regression models. Use of classification approaches (clustering) mighe be useful to understand better the data.

-While the authors discussed the differences in anxiety revelaed by the data, a thorugh discussion of how these factors may influenece the spread of the disease is missing. Please see (Stress as a meaningful variable in models of covid-19 spreading. https://doi.org/10.31234/osf.io/kcpqm )

 -Please state as limitation the fact that is a cross sectional study, given that a longitudinal study would be more meaningful to assess the state of participants after a period of time.

Author Response

Response to Reviewer 2 Comments

Thanks a lot for your critical and kind recognition and suggestions on my manuscripts. Here are my modification and reply according to your revision advice, provided the revised manuscript with tracked changes in red.

Point 1: Please provide a copy of the survey administered. It is unclear how many items composed the survey, how the authors evaluated the state of anxiety and behavioral status. It is unclear whether well-known psychological questionnaires were used for assessment of such aspects. If not this represents a major limitation of the study.

Response: Thank you for your kind and critical suggestions. I’m sorry no to clarify the details of the investigation.

Firstly, I have uploaded the copy of the questionnaire administered in the enclosure, which provides details and specific questions of the survey.

Secondly, I am so sorry no to clarify the details of the details of method of assessing anxiety, Thus, it has been done in the revised in the section of 2.2. Data collection of the manuscript, which is as follow: “And, the Chinese version of Generalized Anxiety Disorder tool (GAD-7) was applied to assess anxiety status and a GAD-7 score ≥10 was considered as a reasonable cutoff point to screen and identify clinical anxiety cases[22]” The GAD-7 is a well-known psychological questionnaire which is widely used for assessment of anxiety. (Page 3, Line 134-139)

Thirdly, I’m sorry not to explain clearly how to assess preventive behavior. Considering that there is no professional evaluation standard for the level in the existing literature, author evaluated the preventive behavior status by means of the answers of related questions. The evaluation standard for that has been done in the revised in the 2.2. Data collection part of the manuscript, which is as follow: “All behaviors (including “Canceling hanging out with friends”, "Wearing a mask", "A significant increasing in the frequency of hand washing", "Giving up going out to where crowds congregate", "Dropping the plan of home-returning or travel during Spring festival", “Calling off family outings and visiting activities”) were performed, then defined the behavior status as positive; as long as there is one behavior unexecuted, defined the behavior status as negative.” (Page 3, Line 117-123)

Reference:[22]Xu, Q.; Mao, Z.; Wei, D.; Liu, P.; Fan, K.; Wang, J.; Wang, X.; Lou, X.; Lin, H.; Wang, C.; et al. Prevalence and risk factors for anxiety symptoms during the outbreak of COVID-19: A large survey among 373216 junior and senior high school students in China. Journal of affective disorders 2021, 288, 17-22, doi:10.1016/j.jad.2021.03.080

Point 2: Are all responders having the same cultural background? Please clarify.

Response: Thanks for your kind and critical guidance and suggestion. I’m sorry not to explain clearly the details of participants. All participants in the survey are college students registered in Henan Province, China. It may be considered that their education or cultural background is similar.

Point 3: Is there any information about effect of financial issues of responders included in the survey?

Response: Thanks for your kind and critical guidance and suggestion. I’m sorry not to explain clearly the details of participants. There is no information about financial issues of responders included in the survey or the questionnaire, which might be taken into consideration in the future.

.

Point 4: Please state if the data gathered by the survey will be made publically available to other researchers.

Response: Thanks for your kind and targeted advice to illustrate the utilization of the data of this investigation. I’m sorry not to explain clearly this. I would like to clarify that  the data are available from the corresponding author on reasonable request.

Point 5: Please specify the independent and dependent variables in the logistic regression models. Use of classification approaches (clustering) might be useful to understand better the data.

Response: Thanks for your kind and targeted guidance and suggestion.

Firstly, I’m sorry not to explain clearly the details of the logistic regression models. The independent variables includes sex, age, place of residence, the cognition about COVID-19, information sources and access, preventive behavior, mental state (worry, fear, anxiety) of college students, and dependent variable is the result of whether be in anxiety.

  Secondly, we considered carefully and also agreed with you. The classification approaches (clustering) can interpreter the factors to some extent. We've already done the classification for the purpose of the study in the period of study design and the classification was based on previous research and what we've done.

Point 6: While the authors discussed the differences in anxiety revealed by the data, a thorough discussion of how these factors may influence the spread of the disease is missing. Please see (Stress as a meaningful variable in models of covid-19 spreading. https://doi.org/10.31234/osf.io/kcpqm ).

Response: Thanks for your elaborate and critical guidance and suggestion. I’m sorry not to take the possible mechanism into consideration. I referred to contemporaneous surveys and literatures on mental health, especially the article that you recommended, and made the following modifications and additions in the 4.Discusion part of the manuscript, which is as follow: “Psychological and physiological mechanisms are activated under COVID-19 epidemic threat, such as eye blink rates, breathing patterns and humoral factors which indicate the stress status could fluctuate [44]. The incidence of stress-related factors raised along with the development of COVID-19 to some extent, which can contribute to mental health disorders. It could be inferred that the level of cognition of COVID-19 and preventing behaviors might influenced the incidence of anxiety by means of having an effect on stress-related factors. ” (Page 11, Line 330-336)

Reference: [44] Weckesser, L.J.; Dietz, F.; Schmidt, K.; Grass, J.; Kirschbaum, C.; Miller, R. The psychometric properties and temporal dynamics of subjective stress, retrospectively assessed by different informants and questionnaires, and hair cortisol concentrations. Scientific reports 2019, 9, 1098, doi:10.1038/s41598-018-37526-2.

Point 7: Please state as limitation the fact that is a cross sectional study, given that a longitudinal study would be more meaningful to assess the state of participants after a period of time.

Response: Thanks for your kind and critical guidance and suggestion. It is really inferior to longitudinal in assessment the state of participants after a period of time. And modification has been done and listed it in in the limitation in the section of 4. Discussion of the manuscript, which is as follow in the revised version: “Finally, the study was designed as a cross-sectional survey, which does not establish a causal hypothesis.” (Page 11, Line 351-352)

Once again I would like to express my gratitude for your kind and elaborate criticism and suggestions to improve the capital of this manuscript. Thanks so much and wish you all the best.

Reviewer 3 Report

This manuscript addresses a very important topic, giving useful information about anxiety symptoms in college students during the Covid-19 pandemic.

The paper is focused, specific and generally well-written. The sample size is large.

I only have a few suggestions that could improve the article in my opinion.

The introduction section, as I said, is very focused and I particularly appreciate its specificity and clarity. I only suggest to help the reader with a brief description of other relevant psychopathological symptoms that have been associated with the pandemic in other studies (e.g. depression, withrawn, aggression, etc.). This paper focuses on anxiety, therefore I suggest to keep this additional part very brief.

Please, add item examples in the description of the measure.

Please, clarify why subject who took too much time in responding were excluded (I agree with this choice but I think it should be justified).

Please, add to the limitation section the fact that comorbility has not been assessed, nor the authors have information on possible co-occurrent and non-pandemic-related events that could have caused the development of anxiety in the subjects.

Author Response

Response to Reviewer 3 Comments

Thanks a lot for your critical and kind recognition and suggestions on my manuscripts. Here are my modification and reply according to your revision advice, provided the revised manuscript with tracked changes in red.

Point 1: I only suggest to help the reader with a brief description of other relevant psychopathological symptoms that have been associated with the pandemic in other studies (e.g. depression, withdrawn, aggression, etc.)

Response: Thank you for your kind and critical suggestions. I’m sorry not to explain clearly the overall mental health condition including other relevant psychopathological symptoms associated with the pandemic. I referred to previous surveys and made the following modifications and additions in 1. Introduction part, which reads: “COVID-19 epidemic has had a significant socio-psychological impact on common people and triggered a wide variety of psychological problems which mainly concentrated in anxiety, panic disorder, depression, obsessive-compulsive, hostility, psychoticism, PTSD and so on.(1, 2)” (Page 2, Line 58-61)

Point 2: Please, add item examples in the description of the measure.

Response: Thank you for your kind and critical suggestions. I’m sorry no to clarify the details of the investigation. I have uploaded the copy of the questionnaire administered in the enclosure, which provides details and specific questions of the survey.

Point 3: Please, clarify why subject who took too much time in responding were excluded.

Response: Thank you for your kind and critical suggestions. I’m sorry no to clarify the details of the exclusion criterion. Thus, it has been done in the revised in the 2.1. Study participants of the manuscript, which is as follow: “Participants aged<18 years or aged>25 years or those who took≤100s to fully respond to the questions (n = 1699) were excluded so as to control quality. Totally, 24678 college students in this analysis met the criteria and were included.” (Page 3, Line 105-107). The criteria referred to the studies with similar questionnaire settings, which guaranteed the validity and quality of the survey.(3, 4)

Point 4: Please, add to the limitation section the fact that comorbidity has not been assessed, nor the authors have information on possible co-occurrent and non-pandemic-related events that could have caused the development of anxiety in the subjects

Response: Thank you for your kind and critical suggestions. I’m sorry to neglect the exclusion of the influence of other recent life events caused anxiety and assessment of comorbidity. Authors would adopt your targeted proposal and add it to the limitation section, which is as follow: “Third, the survey did not evaluate the influence of other recent life events caused anxiety and assessment of comorbidity.” (Page 11, Line 349-351)

  Once again I would like to express my gratitude for your kind and elaborate criticism and suggestions to improve the capital of this manuscript. Thanks so much and wish you all the best.

  1. N. Vindegaard, M. E. Benros, COVID-19 pandemic and mental health consequences: Systematic review of the current evidence. Brain, behavior, and immunity 89, 531-542 (2020).
  2. F. Tian et al., Psychological symptoms of ordinary Chinese citizens based on SCL-90 during the level I emergency response to COVID-19. Psychiatry research 288, 112992 (2020).
  3. Q. Xu et al., Prevalence and risk factors for anxiety symptoms during the outbreak of COVID-19: A large survey among 373216 junior and senior high school students in China. Journal of affective disorders 288, 17-22 (2021).
  4. Q. Li et al., Prevalence and factors for anxiety during the coronavirus disease 2019 (COVID-19) epidemic among the teachers in China. Journal of affective disorders 277, 153-158 (2020).

Round 2

Reviewer 2 Report

None